# The Origin and Evolution of Release Factors: Implications for Translation Termination, Ribosome Rescue, and Quality Control Pathways

**DOI:** 10.3390/ijms20081981

**Published:** 2019-04-23

**Authors:** A. Maxwell Burroughs, L Aravind

**Affiliations:** Computational Biology Branch, National Center for Biotechnology Information, National Library of Medicine, National Institutes of Health, Bethesda, MD 20894, USA; burrough@mail.nih.gov

**Keywords:** release factors, translation termination, peptidyl hydrolase

## Abstract

The evolution of release factors catalyzing the hydrolysis of the final peptidyl-tRNA bond and the release of the polypeptide from the ribosome has been a longstanding paradox. While the components of the translation apparatus are generally well-conserved across extant life, structurally unrelated release factor peptidyl hydrolases (RF-PHs) emerged in the stems of the bacterial and archaeo-eukaryotic lineages. We analyze the diversification of RF-PH domains within the broader evolutionary framework of the translation apparatus. Thus, we reconstruct the possible state of translation termination in the Last Universal Common Ancestor with possible tRNA-like terminators. Further, evolutionary trajectories of the several auxiliary release factors in ribosome quality control (RQC) and rescue pathways point to multiple independent solutions to this problem and frequent transfers between superkingdoms including the recently characterized ArfT, which is more widely distributed across life than previously appreciated. The eukaryotic RQC system was pieced together from components with disparate provenance, which include the long-sought-after Vms1/ANKZF1 RF-PH of bacterial origin. We also uncover an under-appreciated evolutionary driver of innovation in rescue pathways: effectors deployed in biological conflicts that target the ribosome. At least three rescue pathways (centered on the prfH/RFH, baeRF-1, and C12orf65 RF-PH domains), were likely innovated in response to such conflicts.

## 1. Introduction

The translation apparatus includes the most conserved cellular proteins and RNAs that provide the strongest evidence for the shared ancestry of all life forms (Figure 1). Along with the highly conserved ribosomal proteins and ribosomal RNAs, the tRNAs and their cognate amino-acyl tRNA synthetases provide the amino acid building blocks for peptide formation (Figure 1). Additionally, the actual ribosomal decoding of mRNA requires conserved protein factors observed at each of the core steps of translation: initiation, elongation, and release (Figure 1). These factors are primarily GTPases, which act as proofreading switches to ensure accurate decoding of the genetic code. Several of these GTPases display strong evidence for an ancient origin prior the Last Universal Common Ancestor (LUCA) [1].

Notably, prior to the LUCA, the GTPases involved in translation had already differentiated into several distinct clades that were since retained through the history of life on earth (Figure 1) [1]. The first of these clades is represented by EF-Tu/EF-1α, which binds the amino acylated (AA)-tRNA, protects the amino acyl group from hydrolysis, and ensures correct matching of the codon with the anticodon [2,3]. The second clade is prototyped by EF-G/EF2, which plays a role in the translocation of the tRNA as the ribosome elongates the nascent peptide [4,5]. The GTPase IF2/eIF5B defines another clade predating the LUCA, which forms a higher order clade with EF-G/EF-2 (Figure 1). However, in this case it appears to have undergone a functional divergence after the split of the bacterial and archaeo-eukaryotic lineages from the LUCA. In bacteria, the primary member of this clade is critical for the recruitment of the Met-tRNAifMet to the ribosome [6], whereas the eukaryotic cognate plays a central role in the bringing together of the two ribosomal subunits [7]. In contrast, in eukaryotes eIF-2γ, which emerged as an offshoot of the EF-Tu/EF-1α clade, it plays the role of recruiting the Met-tRNAiMet to the ribosome to form the pre-initiation complex [8].

The situation with the translation release factors is more complex as there is no release factor GTPase (RF-GTPase) that can be confidently traced back to the LUCA. Rather, there are two distinct RF-GTPases (also known as the Type-II release factors): bRF-3 which emerged as an offshoot of the EF-G/EF-2 clade in bacteria; and eRF-3 which emerged as an offshoot of the EF-Tu/EF-1α clade, only in eukaryotes [1] (Figure 1). Archaea possess no such dedicated RF-GTPase and appear to use EF-1α itself [9,10]. Despite their distinct evolutionary origins, the RF-GTPases of both bacteria and eukaryotes perform a comparable biochemical function; whereas the elongation factors interact with tRNA, the RF-GTPases interact with the second release factor subunit, a peptidyl hydrolase (RF-PH, the Type-I release factors), which is believed to mimic the tRNA [11,12]. Upon recognizing a stop codon, this RF-PH protein is loaded on to the A-site of the ribosome in place of an AA-tRNA by the action of the RF-GTPase, analogous to the role played by EF-Tu/EF-1α in loading the AA-tRNA during elongation [11]. A conserved glutamine in the RF-PH protein directs a water molecule in place of the usual acceptor amino group in the A-site [13,14,15]. Transfer of the peptidyl chain from the tRNA at the P-site to this water molecule by the action of the peptidyl transferase catalytic center in the ribosome large subunit rRNA results in hydrolysis of the peptidyl-tRNA bond and release of the completed polypeptide [16,17,18].

Sequence and structure analyses of the RF-PH subunits of the release factors show an evolutionary dichotomy, which is like that of the RF-GTPases (Figure 1). Despite having a similar catalytic mechanism and a conserved active site glutamine (Figure 2A,B), the RF-PHs of the bacteria (RF-1 and RF-2) and the archaeo-eukaryotic lineage (aeRF-1) are evolutionarily unrelated (Figure 2A–D) [19,20]. In this context, it is also worth noting a comparable dichotomy in the primary ribosome recycling factors that initiate splitting of the large and small subunits after release: In the archaeo-eukaryotic lineage, this process is initiated by the ABC ATPase Rli1/ABCE1 in conjunction with aeRF-1 [21,22]. In contrast, the bacteria initiate recycling via the evolutionarily unrelated ribosome recycling factor (RRF), which instead contains a conserved domain shared with several amino-acyl tRNA synthetases [23].

These phyletic patterns of the RF-PHs, the GTPases that load them on to the ribosome, and the recycling factors, are atypical for components of the core apparatus that perform essential functions in translation. Hence, this suggests that there are key aspects of the early evolution of the translation apparatus that remain poorly understood. First, it remains unclear as to what was the mechanism of peptide release in the LUCA. Second, even though translation termination is an essential and universal process across the translation systems of extant life forms, the lack of universal conservation of its components, unlike those involved in elongation (Figure 1), raises questions about the importance of this step in the ancestral translation systems. Finally, preliminary surveys have indicated that the RFs may show a much greater tendency for paralog formation and concomitant diversification along with trans-superkingdom lateral transfers. This indicates that multiple, distinct evolutionary forces might be operating on these molecules. In this article we systematically survey the RF-PHs and bring to light several hitherto unnoticed or poorly appreciated aspects of their natural history. We synthesize this information to provide plausible explanations and hypotheses relating to the above questions.

## 2. Results and Discussion

### 2.1. The Prior State of Understanding of the Evolution of Peptidyl Hydrolase Release Factors

The general outlines of the evolutionary history of both the archaeo-eukaryotic and bacterial RF-PH (bRF-PH) have been described as follows: the catalytic domain of the aeRF-1s displays a RNase H fold domain (Figure 2A,C) whereas that of the bRF-PHs shows a distinct α + β fold with no relationship to the former (Figure 2B,D). However, both these types of RF-PHs have further convergently evolved to a three-domain architecture (Figure 2E,F). In the archaeo-eukaryotic lineage the central catalytic aeRF-1 domain is sandwiched between the N-terminal stop-codon-recognition domain of the archaeo-eukaryotic lineage and the C-terminal Pelota domain (which often contains a further zinc-ribbon domain inserted into it [24,25,26] (Figure 2E), a widely-distributed domain which binds specific RNA secondary structures [27,28]. In contrast, the bRF-PH catalytic domain is inserted into the core of the bacterial stop-codon-recognition domain [20]. Together, this combined module is fused to the N-terminal α-helical GTPase-interacting domain [29,30] (Figure 2F). The convergence to a comparable three-domain architecture in the RF-PH subunits of both the bacterial and archaeo-eukaryotic lineages has been proposed to have resulted from the selective pressure to adopt structures mimicking the tRNA [24,31,32] (Figure 2E,F).

On the archaeo-eukaryotic side, the aeRF-1 RF-PH domain underwent a duplication early in the common ancestor of the archaeo-eukaryotic lineage, diversifying into two distinct paralogous versions (Figure 3A). The first of these, the bona fide aeRF-1 (hereinafter classical aeRF-1), serves as the peptidyl hydrolase for polypeptide release during regular translation. The regular aeRF-1 recognizes all three stop codons via a single anti-codon binding domain [24] (Figure 3A,B). The second paralog Dom34 plays a specialized role in recycling a ribosome stalled due to impassable secondary structure in the mRNA (no-go) or a defective mRNA lacking a stop codon due to mutation or premature truncation, which can be caused by endogenous processes or introduced by the extrinsic action of RNase effectors (non-stop) [33,34,35,36,37]. Consistent with this function, Dom34 lacks the conserved glutamine residue and is thereby catalytically inactive. Further, in lieu of the N-terminal stop-codon-recognition domain, Dom34 has a distinct β-barrel domain belonging to the superfamily of Sm-like RNA-binding domains [38] (Figure 3C). This N-terminal domain was at one point incorrectly identified as a nuclease, however, subsequent studies have disabused this notion and indicated that it acts in binding the mRNA [33,39]. In eukaryotes, Dom34 is loaded to the A-site by the GTPase Hbs1, a paralog of the release factor GTPase eRF-3 [36,40].

Bacteria possess two paralogous RF-PHs, RF1 and RF2, which are nearly absolutely conserved across bacteria and are responsible for recognizing distinct sets of stop codons [41] (Figure 4A). Additional paralogs of these conserved versions with more specialized roles appear to have emerged subsequently in bacterial evolution. However, their evolutionary trajectories remain poorly understood beyond the observation that eukaryotes also likely acquired this domain early in their evolution [42] (Figure 4). 

While the above provides an overview of the evolutionary history for RF-PH domains developed over the past three decades, recent investigations have considerably expanded the repertoire of known families in unexpected directions. In the following sections we present these findings, placing an emphasis on their evolutionary and functional implications. Additionally, these findings provide new data with a bearing on two long-standing biochemical and evolutionary problems. First, discovery of novel aeRF-1 families increases resolution of the total evolutionary history of the RF-PHs and provides an insight into translation termination events outside of routine translation and the specialized termination events mediated by Dom34. Second, these findings contribute to the continuing elucidation of distinct ribosome quality control pathways, in particular, with relevance for understanding the evolutionary origins of the eukaryotic ribosomal quality control (RQC) system. In this regard, we present a general evolutionary framework for understanding how RQC emerged in eukaryotes and what this implies for potential parallel innovations in the other superkingdoms of Life, a summary of which is provided in Figure 5. 

### 2.2. Novel Members of the aeRF-1 Superfamily: Their Domain Architectures and Conserved Gene Neighborhood Associations

In addition to the classical aeRF-1 and Dom34 families, two novel clades of the superfamily encompassing at least 19 distinct families were identified through the application of sensitive sequence analysis methods (for complete list, see Appendix A). A multiple alignment of representatives of the known and newly identified clades provides an overview of their shared and distinct structure and sequence features; most notably, it allowed us to determine whether they are likely to be catalytically active as in the aeRF-1 family or inactive as in the Dom34 family (Figure 2G). Many of these families also displayed distinct domain architectures with no previous antecedents among the RF-PHs, expanding their architectural diversity beyond previously described three-domain versions (Figure 2E,F and Figure 3B,C). This multiple sequence alignment was further used to perform a phylogenetic analysis of the aeRF-1 superfamily and produce an updated natural classification of the aeRF-1 proteins (Figure 3A).

Even with the discovery of the novel families, these analyses continue to support a fundamental split in the aeRF-1 superfamily between the Dom34-like and the aeRF-1-like clade, which predated the common ancestor of the archaeo-eukaryotic lineage, as conserved representatives of both clades are found in all archaeal and eukaryotic lineages (Figure 3A). The Dom34-like clade is comprised of only the Dom34 proteins of eukaryotes and archaea, several representatives from the Asgardarchaeal lineage group closer to eukaryotic versions than to other archaeal versions, which is consistent with the recent proposal of the derivation of eukaryotes within this archaeal lineage [43].

The two newly discovered clades of aeRF-1 are more closely related to the classical aeRF-1 clade than the Dom34 clade (Figure 2B). Of these, the so-called bacterial aeRF-1 clade (baeRF-1) largely retains the standard domain architecture of the classical aeRF-1, although in many families of this clade, a distinctive insert comprised of two β-strands instead of the zinc ribbon (ZnR) domain is inserted within the Pelota domain. Given that it observed at the same insertion site as the ZnR, it is conceivable that it represents a degenerate ZnR (Figure 3D). In one distinctive baeRF-1 family, the N-terminal stop-codon-binding domain is retained, however, the C-terminal Pelota domain is lost. Three baeRF-1 families have lost both flanking domains, retaining only the core RNase H fold domain (Figure 2A,C and Figure 3E). Across representatives of several baeRF-1 families, a conserved gene neighborhood association with the ribosome hibernation factor (HPF), involved in translational dormancy, is observed suggesting a potential physical and functional association between these proteins [44,45] (Figure 3F, Appendix A). Phylogenetic analyses suggest that the baeRF-1 clade emerged via a single early lateral transfer from archaea to bacteria (Figure 5). This was followed by their diversification and extensive lateral transfer within the bacterial superkingdom along with loss of the flanking domains in certain clades (Figure 3A,D,E). Crucially, the baeRF-1 have never displaced the classical bacterial RF1/2 proteins, suggesting that they have been adapted for a specialized role rather than functioning as the primary RF-PH in those bacteria that possess them.

Additionally, the classical aeRF-1 clade is related to a further clade which is comprised of four families with very distinctive phyletic patterns and domain architectures (Figure 3A). This clade is prototyped by the yeast Vms1 protein, which was initially characterized as interacting with the AAA+ ATPase chaperone CDC48 [46,47]. Accordingly, this clade uniting the four families was named the Vms1-like RF-1 clade (VLRF-1) clade (Figure 2G). The eukaryotic family of this clade is referred to as the eVLRF-1 family and is found in most eukaryotes except the basal-most diplomonad and parabasalid lineages and the apicomplexa. Notably, this eukaryotic family is related to the exclusion of all other families in the VLRF-1 clade to a family found in certain bacteria, predominantly Bacteroidetes and certain heimdallarchaeota (bVLRF-1 family, Figure 3A). Additionally, the VLRF-1 clade contains two other families, which are found, respectively, in a subset of the archaea and some bacteria, primarily of the actinobacterial and chloroflexi lineages. The archaeal VLRF-1 family (aVLRF-1; Figure 2), while lacking the C-terminal Pelota domain of the classical aeRF-1 clade, has a long coiled-coil extension N-terminal to the stop-codon-binding domain (Figure 3G). The actinobacterial/chloroflexi VLRF-1 family (acVLRF-1, Figure 2) has lost the stop-codon-binding domain and is instead fused to a distinctive short N-terminal extension (Figure 3G).

Phylogenetic analysis and domain architectural patterns indicate that the VLRF-1 clade emerged within archaea and was then transferred to bacteria independently of the transfer that spawned the baeRF-1 clade (Figure 3A,G,H and Figure 5A,B). Strikingly, this analysis indicates that the eukaryotic VLRF-1 family was not directly derived from the aVLRF-1 family, to which they are only distantly-related, but rather from the bVLRF-1 family (Figure 3A and Figure 5A). This conclusion is also supported by the sequence conservation shared by the bVLRF-1 and eVLRF-1 in the N-terminal extension that is only found in the bacterial families of the VLRF-1 clade (Figure 3A,G,H). However, members of the eukaryotic clade have considerably diversified in domain architecture. In early-branching eukaryotes, like kinetoplastids, there is an extension to the ancestral bVLRF-1 core either in the form of an α-helical N-terminal domain seen in kinetoplastids, or a F-box domain which is seen in the heterolobosean Naegleria and some ciliates (Figure 3H). In the common ancestor of stramenopiles, plants, amoebozoans, animals, and fungi, the core was embellished by the accretion of a number of other domains: (1) a N-terminal Rei1-like C2H2-Zn-finger (Rei1-ZnF); (2) a C-terminal run of 3 ankyrin repeats; (3) novel treble-clef fold Zn-binding domain which we name the a Vms1-treble-clef (VTC) domain; and (4) VIM, a motif which has been previously shown to be a diagnostic feature of proteins that interact with the AAA+ ATPase chaperon CDC48 [48] (Figure 3H). In some cases, such as the yeast Saccharomyces cerevisiae and related fungi, the VTC domain might be lost, and in stramenopiles the order of the VTC domain and the VIM are inverted (Figure 3H). Of these domains, the Rei1-ZnF is notable in that it is found in the Rei1 protein involved in eukaryotic ribosome assembly. In Rei1, the Rei1-ZnF domain likely binds the ribosome large-subunit near the eL23 protein, while a C-terminal helical region blocks the ribosome exit channel to keep it protected prior to actual use in the cytoplasm [49] (Figure 3I). The Rei1-ZnF is also found in some previously uncharacterized eukaryotic proteins, where it is fused to the C-terminus of the Shwachman-Bodian-Diamond syndrome (SBDS) domain involved in the late cytoplasmic assembly of the eukaryotic 60 ribosomal subunit by removal of eIF6 (Figure 3I) [50,51]. These features strongly suggest that the eukaryotic VLRF-1 proteins are likely to associate with the ribosome via their N-terminal Rei1-finger domain.

### 2.3. Members of the aeRF-1 Superfamily Display Sequence and Structure Diversity in Their Core RNase H Fold Domain 

The RF-PH catalytic domain shared by all aeRF-1s belongs to the RNase H fold, which encompasses a vast assemblage of protein domains descending from a common ancestor (Figure 2A,C). These domains can be classified into multiple distinct clades, each sharing their own functional attributes. The most widely known of these clades is the titular Ribonuclease H-like superfamily, which contains various distinct families functioning as RNA and DNA endo- and exonucleases. Another distinct assemblage with this fold, typically with two tandem RNase H fold domains, possess NTPase activity [52] and act as cytoskeletal proteins (actin/FtsA) or NTP-dependent chaperones (e.g., HSP70). Related families within this fold also function as accessory scaffolding domains for other enzymes including nitrogenase, creatinase/prolidase, and DNA methyltransferases [53,54,55]. However, an analysis of the families of domains sharing RNase H domain reveals that the earliest branching assemblage of the RNase H fold is comprised of the catalytically inactive versions found in the conserved ribosomal proteins such as L18 and S11. A key active site residue, typically an aspartate or glutamate, found in the first strand (strand-1) of the RNase H domain separates the nuclease/NTPase versions from non-enzymatic versions [56,57,58]. As previously mentioned, the ribosomal versions lack the catalytic residue that plays the RNA-binding/structural role in the ribosome. The RNase H domain in the aeRF-1 superfamily is specifically related to these ribosomal proteins to the exclusion of all other RNase H fold domains. Consistent with this, the aeRF-1 RNase H domain lacks the aforesaid active site residue in the strand-1 typical of NTPases/nucleases of this fold (Figure 2C). Instead, it has an extended loop, which bears the catalytic glutamine, connecting the third of the conserved β-strands of the fold to the downstream α-helix. The catalytic glutamine is deeply inserted into the A-site to come close to the peptidyl transferase ribozyme active site, and as stated above, directs a water molecule to this active site [24,25] (Figure 2A,C). 

This structural and evolutionary framework of the RNase H fold helps to glean functional insights regarding the newly identified aeRF-1 superfamily clades. First, analysis of the multiple sequence alignments of the new aeRF-1 clades (Figure 2G) allows identification of versions which might be catalytically active like the classic aeRF-1 as opposed to being inactive like Dom34 on the basis of the catalytic glutamine in the above-mentioned loop (Figure 2A,C). Additionally, aeRF-1 superfamily members might contain one or more conserved arginines at the N-terminus of the helix following the three consecutive N-terminal β-strands of the RNase H fold (Figure 2A,G). These residues serve to bind the large subunit rRNA and have also been implicated in interaction of eRF-1 with the cognate GTPase RF (eRF-3) as a tRNA-mimic (see below) [24,59]. 

Inspection of the multiple sequence alignments shows that the VLRF-1 clade contains a comparable loop with a conserved Q as in the catalytically active classic aeRF-1s, as well as several conserved arginine residues in the helix downstream of the core β-strands (Figure 2G). Further, on account of their three-domain architecture, the classic aeRF-1 and Dom34 families are believed to function as tRNA mimics that interact with the cognate RF GTPase, in a similar way to the AA-tRNA interacting with EF-Tu/EF-1α [11,32]. In eukaryotes, at least, VLRF-1 RF-PH domains have acquired flanking domains that could confer a multi-domain architecture that could approximate a tRNA mimic-like structure. Hence, they could in principle interact with a cognate GTPase, however, the identity of such a GTPase remains unclear and at least some circumstantial evidence suggests that GTPase activity is not coupled to peptidyl hydrolase activity in the eukaryotic VLRF-1 family [60]. Instead, the domains fused to the eukaryotic VLRF-1 family could be involved in an interaction with the ribosomal proteins, particularly the above-mentioned Rei1-ZnF domain (Figure 3H), in addition to the VIM motif interacting with the ATPase chaperone CDC48 [48]. 

Recently, it has been asserted that the lack of a conserved di-glycine amino acid pair in the eukaryotic VLRF-1 family, which is an absolutely conserved, convergently acquired feature of both the classical aeRF-1 family and the bRF-PH domains, calls into question its previously demonstrated function as a peptidyl-hydrolase [61]. This reasoning was used as justification for claiming the alternative RNase activity for the domain. However, this assertion relies on the supposition that the di-glycine amino acid pair should be required for peptidyl-hydrolase enzymatic activity. This is a position unsupported by the balance of experimental evidence which has converged on an understanding that the motif supplies flexibility to the RF-PH catalytic loop to ensure proper positioning of the catalytic glutamine in the active site pocket [15,62]. The above claim also overlooks existing small residue conservation in both the eukaryotic VLRF-1 and bVLRF-1 families: a single absolutely conserved glycine residue is always coupled with a near absolutely conserved tiny (typically either a serine or alanine) residue immediately upstream of the glutamine residue to form a GxuQ motif (where “u” is a tiny residue), a conservation pattern very much comparable to known RF-PH domains. Indeed, the newly discovered families can be viewed as the continuation of the spectrum of small residue conservation seen in proximity of the catalytic glutamine residue. For example, the classical aeRF-1 domains carry a GGQ motif, the bacterial RF domains carry a uxGxGGQ motif, and both the aVLRF-1 and acVLRF-1 families carry a GGxSQ motif (Figure 2G). Additionally, the VLRF-1 family has a longer catalytic loop than its classical aeRF-1 counterpart (Appendix A), a feature that could accommodate the necessary flexibility needed to access the peptidyl transferase active site, perhaps in combination with the absence of the small subunit in the rescue context. 

Shannon entropy plots for the sequence alignments of the broadly disseminated families in the baeRF-1 clade (e.g., families 1,2,6), as compared with the other families, indicate poor conservation in the region at and surrounding the catalytic loop, with punctuated exceptions in the downstream helix largely representing conservation of specific arginine residues (Appendix A). Unlike the classical aeRF-1 and VLRF-1 clades, most families in the baeRF-1 clade, particularly those versions which have lost one or both the flanking RNA-binding domains, usually appear to lack the conserved glutamine residue in the catalytic loop. This, together with the divergence in this region, suggests that they are catalytically inactive as peptidyl hydrolases. However, a few baeRF-1 families, perhaps most notably family 10 (Figure 2G and Figure 3A), do contain a well-conserved glutamine residue typical of the active families and could potentially function as peptidyl hydrolases. Further, an analysis of the distribution of the lengths of the catalytic loop of the baeRF-1 clade families revealed a striking lack of constraint on its length, notably in the families lacking the catalytic glutamine (Appendix A). This pattern sharply contrasts with observations in families that contain the conserved glutamine and are known to or predicted to catalyze peptidyl hydrolase reactions. While these loop lengths can be different between these families, they are confined to a narrow range within any given family. The conservation of arginines or otherwise positively charged residues in the helix downstream of the catalytic loop (Figure 2A,G), even in versions lacking fusions to the codon recognition and pelota domains (Figure 3E), suggests that representatives of the baeRF-1 clade broadly interact with the LSU rRNA and likely insert into the A-site. 

These observations regarding most of the baeRF-1 families may be compared to the catalytically inactive Dom34 family and all share poor conservation in the catalytic loop and absence of the catalytic glutamine (Figure 2G). Further, the above-noted, persistent operonic association with the ribosome hibernation factor family of proteins (Figure 3F) suggests that these families might typically interact with the ribosome in conjunction with these factors. These features suggest that they might function as non-catalytic negative regulators of translation, which insert into the A-site in lieu of a tRNA to facilitate ribosomal inhibition or disassembly by inserting into the A-site. However, the few families conserving the catalytic glutamine might function as alternative RF-PHs that are used in special circumstances, much like the eukaryotic VLRFs.

### 2.4. Diversity and Phylogeny of bRF-PHs in the Bacteria

The second type of RF-PH, i.e., bRF-PHs, have been noted previously to exist as a range of paralogous versions, however, the totality of this diversity and its evolutionary implications has not been described. In domain architectural terms, the canonical bRF-PHs contain an N-terminal helical GTPase-interacting domain which is followed by the stop-codon-recognition domain, into which the RF-PH domain with the convergently evolved catalytic glutamine has been inserted (Figure 2F and Figure 4B). As noted above, most bacterial genomes, excepting severely reduced genomes such as in some tenericutes, contain two copies of this version (RF1 and RF2), which specialize in different sets of stop codons during general translation termination (Figure 4A). This suggests that both RF1 and RF2 were likely present in the common ancestor of all bacteria and arose via duplication in the stem lineage of all extent bacteria.

Two additional versions of the bRF-PH, both of which retain the catalytic glutamine residue (Figure 2B,D), are widespread in bacteria. The first of these is the ArfB/yaeJ, which contains a standalone RF-PH domain connected by an extended linker to a C-terminal α-helical region (Figure 4C). This helical segment localizes to the incompletely occupied codon recognition center of ribosomes that have been “jammed” by non-stop translation events [63,64]. Thus, ArfB functions as a single-component bacterial ribosome-rescue factor. ArfB is widespread in bacteria but is sporadically absent in the bacteroidetes and the actinobacteria, and missing entirely in other lineages including the firmicutes, fusobacteria, thermatogae, aquificae, and thermus-deinococci (Figure 5B). Despite these absences, the phyletic pattern of ArfB is suggestive of a trajectory wherein ArfB emerged early in bacterial evolution and has been subsequently lost in some lineages, possibly due to the presence of other functionally overlapping rescue pathways in bacteria [65,66] (see below, Figure 5A). Being a single-component rescue pathway, ArfB does not show extensive conserved gene neighborhood association beyond a persistent association with a pseudouridine (PSU) synthase (PSYN) gene across several proteobacterial classes (Figure 4D, Appendix A). Strikingly, this PSYN appears most closely-related to the 23S rRNA-modifying PSYN family which includes RluC and RluD [67], suggesting a coupling of non-stop rescue to either the PSU modification of the peptidyl transferase ribozyme active site or the mRNAs themselves [68].

The other bacterial Rf-PH that has been relatively poorly studied is the prfH or RFH release factor. This protein lacks the N-terminal GTPase-interacting domain, and in contrast to ArfB/yaeJ, retains a version of the anticodon recognition domain [69] (Figure 4E). It was originally reported as showing a conserved gene neighborhood association with the then functionally uncharacterized RtcB protein (Figure 5F) [69], which was subsequently demonstrated to function as the RNA ligase [70,71,72]. A subsequent analysis classified this RFH-RtcB association as a representative of a diverse and vast collection of systems which repair RNA in response to damage caused by toxin effectors deployed in the biological conflicts between invasive elements like plasmids, viruses, and conjugative elements and their host cells [73]. In some cases, as is typical of many counter-conflict systems, further accretion of accessory genes around the core two-gene system (RFH-RtcB) is observed (see Appendix A for complete list): (1) A gene coding for a TPR domain, which could serve as a scaffold for other components comparable to the TPR protein Rqc1 in eukaryotic ribosome rescue (see below, Figure 4G–I); (2) A gene encoding a previously uncharacterized member of the Polβ superfamily, members which catalyze nucleotidyltransferase (NTase) reactions (Figure 4G,I). Another member of the polβ superfamily, the RlaP NTase, is found across several distinct counter-RNAse repair systems [73]. While this RFH-associated NTase does not appear to be specifically related to the RlaPs, it possibly plays a biochemically equivalent role in repairing RNAs like tRNAs that may have undergone further exonucleolytic degradation after an initial endonucleolytic cleavage event [73,74]; (3) A protein combining one or two ZnR domains with a domain of unknown provenance containing a highly positively charged helical region at the C-terminus (Figure 4H,I); (4) A protein again lacking any detectable homologs but possessing highly charged helical regions (Figure 4I). These latter two proteins could interact with rRNA either via the N-terminal ZnF domains or through their various charged regions. One further possibility is that one or more of these positively charged helical regions could function as charged “tails” comparable to those found in ArfB and other bacterial ribosome rescue factors (see below).

Thus, contextual evidence from conserved gene neighborhoods strongly supports a role for prfH/RFH as a release factor that is specifically coupled to RNA-repair by the RtcB ligase. This contrasts with ArfB which functions in an “institutionalized” rescue pathway responding to stalled ribosomes arising from intrinsic transcription and translational failures. Specifically, the prfH/RFH system likely functions in rescue necessitated by impairment of the ribosome by the action of extrinsic attacks from RNase effectors [73], where it hydrolyzes the peptidyl-tRNA even as the associated RtcB ligase repairs cleaved RNAs that jam the ribosome. Accordingly, the prfH/RFH-RtcB operonic unit is sporadic in its phyletic distribution, indicating dissemination via horizontal gene transfer (HGT) (Figure 5B, Appendix A). 

### 2.5. bRF-PH Domains Transferred to Eukaryotes

Just as the RF-PH domains of the aeRF-1 superfamily have been transferred to bacteria, bRF-PH domains have also been transferred to eukaryotes. The provenance and distribution of these RF-PHs has been previously studied by Huynen et al. [42] who note two release factors acquired by eukaryotes that retain the ancestral architecture, respectively, descending from the bacterial RF1 and RF2 (Figure 4A). These are well-conserved across most eukaryotes with functional mitochondria and appear to operate within a similar stop-codon-recognition dichotomy as their bacterial progenitors. They have occasionally given rise to further lineage-specific versions [42]. Additional bRF-PHs are also prevalent in eukaryotic lineages which acquired photosynthetic plastid organelles [42].

However, another bRF-PH transferred to eukaryotes, ICT1, is highly conserved in eukaryotes. Strikingly, this version is clearly related to the ArfB/yaeJ family of bacterial ribosome rescue release factors and retains the C-terminal helical region peculiar to these bRF-PHs (Figure 4C). It has been recently demonstrated to perform an equivalent role as ArfB/yaeJ with respect to the mitochondrial ribosomes [75]. A further version, the C12orf65 protein has a parallel architecture combining the RF-PH domain with a C-terminal positively charged helical region. While this RF-PH remains experimentally uncharacterized, it has been linked to several mitochondrial deficiencies [76,77,78,79] and is speculated to play an overlapping role with ICT1 in organellar ribosomal rescue pathways [77]. Intriguingly, we identify a fusion between C12orf65-like proteins and the MJ1316 domain which recognizes RNA 2′-3′ cyclic nucleotide ends [73,80] (Figure 4C). The MJ1316 domain displays diverse fusions across eukaryotic lineages to domains involved in RNA repair [73], suggesting that the release factor activity of C12orf65 might be potentially coupled with repair of organellar RNA damage as proposed above for the RFH/prfH system [73]. The relative rates of evolution of the two proteins, with ICT1 being more conserved than c12orf65, could be interpreted as supporting a scenario where it operates along with RNA repair, specifically occurring within the context of biological conflicts.

The current genomic data allows detection of C12orf65 homologs in several additional lineages beyond those previously reported [42], including the basally positioned kinetoplastids. Therefore, the current data points to more similar phyletic patterns for ICT1 and C12orf65 than was apparent before. However, we observe that the ArfB family as currently defined contains a previously unidentified structural dichotomy. Specifically, while several members hew close to the observed structure of the canonical RF1/RF2 bRF-PHs catalytic domains (Figure 2B,D), a large subset contains a helical segment (previously labeled as the αi segment [81,82]) between the final two β-strands of the core β-sheet of the bRF-PH domain (Figure 2B). Hereafter, we refer to these versions as ArfB class 1, with the more ancestral versions referred to as the classical ArfB. Strikingly, comparison of known structures to multiple sequence alignments reveals that the ICT1 RF-PH domain clearly contains this αi segment, suggesting direct descent from the ArfB class 1 while C12orf65 lacks the segment. Phylogenetic analysis further supports this distinction, clearly grouping ICT1 with ArfB class 1 and C12orf65 with classical ArfB domains (Figure 4A). This strongly supports a scenario where the eukaryotes acquired two distinct bRF-PHs early in their evolution, likely from the independent transfer of two distinct antecedents of the ArfB family (Figure 4A and Figure 5A). 

A final, previously unreported version of the bRF-PH domain in eukaryotes is prototyped by a version found in the mitochondrial ribosomal protein S35 (MRPS35). This version lacks the GTPase-interacting and codon-recognition domains, in a similar way to ICT1 and c12orf65 versions. However, MRPS35 has acquired two distinctive helical domains flanking the core bRF-PH domain (Figure 4J, Appendix A for complete list). Unlike ICT1 and C12orf65, MRPS35 is enzymatically inactive, as it lacks the conserved glutamine in the catalytic loop. Despite the considerable sequence divergence between MRPS35 and other characterized bRF-PHs, the presence of the αi helical segment (Figure 2B) suggests an origin via duplication of ICT1 (Figure 4A), with subsequent loss of the catalytic glutamine residue and recruitment for a structural role in the mitochondrial ribosome [83,84]. It is also possible that MRPS35 could function in some non-catalytic rescue role, perhaps approximating the role of Dom34 in the archaeo-eukaryotic lineage in triggering ribosomal subunit disassociation. Along these lines, it is interesting to note that ICT1 similarly has been observed as associating with the large subunit of the mitochondrial ribosome [85,86,87].

## 3. Functional and Evolutionary Implications

### 3.1. The State of Translation Termination in the LUCA and the Early History of RF-PH Domains

The observation that evolutionarily unrelated RF-PHs emerged early in the “stems” of both the bacterial and archaeo-eukaryotic lineages, together with the recruitment of distinct versions of cognate RF-GTPases, has long stood as an evolutionary puzzle. The solution to this paradox has been to either speculate that one of the types of extant RF-PHs was present in the LUCA and displaced in one or the other lineage, or that a different factor was ancestrally responsible for peptide release, with each of the two lineages independently evolving their own solution that displaced the ancestral mechanism (Figure 1). 

On the surface, the demonstration in recent years that both the aeRF-1 and bRF-PH superfamilies have extensive diversity in all three superkingdoms of Life would seem to argue for the first scenario. Under this scenario these newly uncovered radiations potentially represent remnants of the ancestral RF-PH in opposing lineages. However, phylogenetic and domain architectural evidence places the ultimate provenance of these newly described release factors squarely within either the archaeo-eukaryotic (aeRF-1 RF-PHs) or the bacterial lineage (bRF-PHs). In the case of the baeRF-1 clade the diversification in the bacteria likely occurred after a single transfer of a classical aeRF-1 version from the archaeo-eukaryotic lineage to a bacterium early in their evolution followed by dissemination via extensive HGT within bacteria (Figure 5A). All characterized bRF-PHs in eukaryotes have predominantly organellar functions and are rooted within the bacterial radiation of these domains. This strongly suggests that they were all acquired by the stem during the primary endosymbiotic event that defined eukaryogenesis or from other early endosymbionts. Subsequently, they underwent diversification early in eukaryotic evolution (Figure 5A). Together, these observations instead point in the direction of the second hypothesis where no conserved RF-PH was inherited from the LUCA. 

Several lines of evidence support a form of the second scenario: first, the observation that the stop signal is also a base triplet, like the codons, used to encode the amino acids suggests that in the ancestral organism the stop was treated similar to the regular codons which specify amino acids. Second, the same stop codons are used in both the archaeo-eukaryotic and bacterial divisions. This indicates that the same stop signals were already present and interpreted in the LUCA. While currently these stop codons are decoded by protein RF-PHs, in both the archaeo-eukaryotic and bacterial divisions of life, the primary versions of the RF-PHs in the two divisions of life are evolutionarily unrelated. However, they notably utilize a biochemically equivalent glutamine in conjunction with the universal peptidyl transferase active site to catalyze peptide release. Considering these observations together, a plausible scenario for the release factor in the LUCA is the presence of a modified tRNA-like molecule (Figure 1). It is conceivable that this tRNA-like molecule presented a water molecule to the peptidyl transferase active site using a glutamine-like moiety attached to it. This is consistent with the class of hypotheses which favor a larger role for RNAs in the early stages of life [88,89,90]. In this regard, it is also notable that the bacteria and the archaeo-eukaryotic lineages show differences in ribosome recycling which follows termination. Interestingly, the recycling factor specific to the bacterial lineage, RRF, is related to the non-catalytic tRNA-binding domain of arginyl tRNA synthetases and the catalytic editing domain of the alanyl and threonyl tRNA synthetases [91]. Indeed, RRF has been observed to bind RNA in a similar way to both domains of aatRs [92,93]. This relationship might again favor the ancestral presence of a dedicated tRNA in peptidyl chain release, with RRF being derived from an RNA-binding protein which perhaps bound such a tRNA in the ancient translation system (Figure 1).

Under the above scenario this function of the ancestral tRNA-like molecule would have been independently taken over by protein factors only after the divergence of the bacterial and the archaeo-eukaryotic lineages. This was perhaps favored due to the ability of a protein to allow unambiguous decoding of the stop codons along with resistance to misreading by other amino-acylated tRNAs that resembled the original tRNA-like stop codon decoder. Considering this proposal, it is tempting to suggest that the tmRNA from the tmRNA-smpB system (Figure 5B) in bacteria might secondarily resemble some aspects of this proposed termination mechanism in LUCA (Figure 1). It is conceivable that this mechanism survived in the context of an ancient rescue pathway in the stem of the bacterial lineage. The simultaneous coupling of this pathway to the degradation of unfinished polypeptide chains could have in turn contributed to the failure of complete prokaryotic ubiquitin systems to gain a foothold as a core process in the bacteria [94,95,96,97].

### 3.2. Multiple Early Recruitments of Paralogous RF-PHs to Distinct Ribosome Rescue Pathways 

Another important point that our survey of the RF-PHs raises is that multiple RF-PH-like proteins can coexist alongside the primary RF-PHs in the same organism without displacing the latter. Moreover, RF-PHs from either division of life (bacterial or archaeo-eukaryotic) can function in the other. In the case of the aeRF-1 superfamily it is apparent that RNase H fold RF-PH domain can function in the A-sites of both archaeo-eukaryotic and bacterial-type ribosomes. It remains to be seen if the same might apply for bRF-PH domains, like those of ICT1 or c12orf65. Further, this wide adoption of diverse RF-PH paralogs points to their utility in contexts that require peptide release or translation termination beyond the regular recognition of stop codons. These contexts involve either relief from endogenous ribosomal stalling events, which can harm the cell through wasted resource allocation or production of misfolded proteins that can form aggregates, or relief from “ribosome-jamming” arising from the action of effectors on the translation apparatus for killing the cell or inducing stasis. Recruitment of RF-PH paralogs to the pathways which combat these insults was thus driven by the benefits accrued from relief of ribosome stalling. Particularly in the latter contexts, RF-PHs would need repeated innovation to cope with new translation-targeting effectors that emerge as part of escalating arms races in biological conflicts [98,99].

An example of such duplication followed by fixation occurred early in the archaeo-eukaryotic lineage with the emergence of the catalytically inactive Dom34 clade, which appears to have specialized in relieving stalled ribosomes due to a variety of reasons such as the lack of stop codons in aborted or otherwise truncated transcripts, or the presence of a blocking secondary structure on the mRNA. Rather than active hydrolytic release of the nascent peptide from the tRNA, this factor simply acted to separate the ribosomal subunits and terminate translation of the defective transcript [66,100]. 

Comparably in bacteria, rescue solutions including the tmRNA/SmpB and ArfA systems emerged largely independently of a secondary peptidyl hydrolase activity [65,66]. These proteins harbor charged helical regions that recognize non-stop translational stalling events via interaction with exposed regions in the anticodon recognition site. In this regard it is has been recently reported that the ArfT protein in *Francisella tularensis* with a comparable charged α-helix can similarly recruit the canonical bRF-PHs, RF1/RF2, to stalled ribosomes [101]. We observed that rather than being a *Francisella*-specific novelty, the ArfT protein is a single representative of a broadly distributed family with sporadic representation in several bacterial lineages, and even certain archaea and eukaryotes such as fungi and plants (Figure 5B, Appendix A). Several of these newly recognized bacterial and plant ArfT proteins had been previously implicated in stress response, consistent with a role in ribosome rescue [102,103,104]. The ArfT α-helix implicated in direct ribosome interaction is often found in multiple copies. ArfT-like proteins were observed containing arrays between one to eight such repeats (Figure 4K, Appendix A). Thus, such mechanisms of recognizing stalled ribosomes could also proceed via the recruitment of general RF-PHs and serve as an alternative mechanism to rescue stalled ribosomes through direct hydrolysis of the peptidyl tRNA. This niche of ribosome rescue via catalytic action appears to have been further colonized with the emergence of dedicated active paralogs of the general bRF-PHs early in bacterial evolution in the form ArfB/YaeJ. This pathway has been retained in the mitochondrial ribosomes of eukaryotes centered on the ArfB/yaeJ homolog ICT1 [75].

A further context for the recruitment of RF-PHs is indicated by the bacterial RFH/prfH, which appears to be coupled with RNA repair. This potentially acts to release peptides from ribosomes which have been stalled due to an attack on tRNA or other RNAs by the action endoRNases that are widely deployed in biological conflicts [73]. Growing evidence points to the pervasiveness of effectors deployed in biological conflicts that “jam” the ribosome [73], i.e., cause translational stalling by targeting tRNA, rRNA, mRNA at the ribosome or in some cases ribosomal proteins [105,106,107]. Moreover, there is also strong support for the evolution of counter-strategies that help alleviate such blocks via repair of the damaged RNAs. Thus, the RFH/prfH family’s action is potentially coupled with the repair of the damaged RNA by the linked RtcB RNA ligase (Figure 4F–I) [73]. While largely experimentally uncharacterized, another such bRF-PH, C12orf65 from eukaryotes, might play a similar role in the context of RNA damage (Figure 4C).

Considering the above, we posit that the widely distributed paralogous versions of the aeRF-1 superfamily in bacteria (baeRF-1 family) are also likely to play a role in comparable processes by relieving stalled ribosomes either via a non-catalytic mechanism like Dom34, or in some cases by catalyzing the release of the stalled peptide. Consistent with this proposal, the baeRF-1 clade is widely disseminated by lateral transfer rather than showing a pattern of predominantly vertical inheritance that is typical of the “institutionalized” classical aeRF-1 and Dom34 versions (Figure 5B). Further, the genomic associations, which we recovered linking diverse baeRF-1 families with the ribosome hibernation factors family (YfiA/HPF) that facilitate formation of 100S and 70S inactive ribosomal aggregates [108,109,110,111], suggest a function linked to translational shutdown (Figure 3F). One possibility is that these aeRF-1 domain proteins block access to the peptidyl transferase active site at the ribosome and subsequently allow the YfiA/HPF-like proteins to lock the ribosomes as inactive aggregates during stress conditions or effector attacks. A wrinkle in this observation is that some baeRF-1 families are found as standalone RF-PH domains on the genome, in other cases they are found in the standard aeRF-1 three-domain arrangement (Figure 1). At least the versions with the full domain complement could directly recognize stop codons to halt translation, perhaps during stress conditions. In either case these mechanisms could preserve ribosomes for later resuscitation [112,113,114,115].

Further cases of cross-kingdom transfer of factors involved in rescue process are seen in the form of sporadic transfers to eukaryotic lineages of both the ArfT and the tmRNA/SmpB. While eukaryotic tmRNA homologs had been previously reported in certain stramenopiles [116,117,118], we identify several of their counterpart SmpB proteins (see Appendix A) and report a further presence in rhodophytes and the crytophyte *Guillardia*. The strongest sequence affinities for these SmpB proteins are with cyanobacteria, consistent with potential transfer during plastid endosymbiosis that gave rise to the original photosynthetic eukaryotes. While this tmRNA/SmpB system appears to have been lost in the plant lineage, it has persisted in some basal chloroplast-containing lineages and was also transferred via secondary endosymbiotic events to further lineages including the stramenopiles [119,120].

### 3.3. Origin of the Eukaryotic Ribosome Quality Control System

The eukaryotes are distinguished from the prokaryotic superkingdoms by possessing an elaborate and dedicated ribosome rescue system in the form of the RQC pathway which rescues stalled translation events. The release factor of this pathway remained unknown until recently [60]. Remarkably, the responsible release factor, Vms1/ANKZF, a member of the VLRF-1 clade of the aeRF-1 superfamily is most closely related specifically to those from the bacteroidetes lineage and Heimdallarchaeota (Figure 3A,G). Heimdallarchaeota are part of the Asgardarchaeal branch and are proposed to be the sister group of eukaryotes. This raises the possibility that VLRF-1 was specifically transferred from bacteroidetes to the common ancestor of Heimdallarchaeota. However, the absence of VLRF1 in basal eukaryotes suggests an alternative scenario. Multiple members of the bacteroidetes lineage are known to form endosymbiotic associations with eukaryotes [121], and early, we presented evidence for certain conserved eukaryotic proteins having their origins in bacteroidetes [99,122]. Hence, eukaryotic Vms1 could be another example of a protein with such an origin, pointing to the key role played by bacterial endosymbionts beyond the mitochondrial progenitor in the emergence of eukaryote-specific systems. In bacteria and archaea, the members of the VLRF-1 clade are standalone versions occurring independently of the other components of the RQC system. Thus, it is possible that they might function comparably to the RF-PH paralogs discussed above that function as single-component release factors in specialized contexts (Figure 5B). Thus, the core of the eukaryotic RQC systems appears to have emerged from the acquisition of such a specialized RF-PH.

The identification of Vms1 as the peptidyl-hydrolase of the RQC system follows the delineation of its interactions with other known components of the pathway, including the AAA+ ATPase CDC48 [48] and Rqc1, Rqc2, Ltn1, and Npl4 [60,123,124]. Of these, Rqc2 is particularly notable. Prior studies have suggested a role for it in the formation of template-independent C-terminal alanine-threonine (CAT) tails by the peptidyl transferase center to facilitate the extrusion of the stalled peptide from the exit channel for subsequent ubiquitination and degradation [125]. Earlier we had shown that this core player of the RQC complex goes back to the LUCA [126] (Figure 5B). Given its universal presence, a possible functional role for the protein in as-yet unknown ribosome rescue pathways in the bacteria and archaea could be fruitful avenues of future experimental investigation. The role of the Rqc2 protein in these kingdoms could center on a process similar to CAT-tailing. Another, not mutually exclusive possibility is that these proteins could be involved in RNA modification reactions during rescue, as we had previously identified a relationship between the N-terminal domain of these proteins (NFACT-N) and the Fpg, MutM, and Nei/EndoVIII family of DNA glycosylases, and shown that several of the residues needed for catalyzing base removal were spatially shared by these two domains. The conserved gene neighborhood association observed between the ArfB and PSYN genes in some bacteria (Figure 4D, Appendix A) again points to potential links between RNA base modifications and ribosome rescue pathways.

Another key aspect of the emergence of the RQC system was its coupling with the expanded ubiquitin system for protein degradation, which is typical of eukaryotes. This happened via the Ltn1 and Npl4 proteins that emerged early in eukaryotic evolution. Our analysis also suggests that the Rqc1 family of proteins, featuring a tetratricopeptide repeat toroidal module, had their origins in bacteria. Thus, the eukaryotic RQC system can now be shown to be a complex innovation involving: an ancient core inherited from the LUCA in the form of Rqc2, an archaeo-eukaryotic innovation in the form of CDC48, and several eukaryote-specific innovations relating to the coupling with the Ub-system (e.g., Npl4 and Ltn1) and key catalytic and structural components of bacterial provenance (e.g., Vms1, Rqc1) (Figure 5A).

### 3.4. Alternative Catalytic Mechanisms for Releasing Peptides from Peptidyl-tRNA

In addition to the previously discussed RF-PHs, there are other enzymes that catalyze the release of amino acids and peptides from tRNA. Such enzymes have emerged in at least two structurally unrelated folds: (1) the bacterial-type D-Tyr-tRNA(Tyr) deacylases with the STAS-like fold and (2) the amino acyl/peptidyl hydrolases of the peptidyl-hydrolase-phosphorylase fold. The latter fold features a key family of such enzymes, universally conserved in bacteria, prototyped by the peptidyl-tRNA hydrolase (Pth) (Figure 5B). The Pth catalyzes the hydrolysis of the peptidyl-tRNA bond in translation products that prematurely dissociate from the ribosome active sites [127,128,129]. It functions in conjunction with the RRF and elongation factor-G [23,130,131] to complete the release of such aborted translation products from the ribosome. The Pth was acquired by eukaryotes from bacteria and is generally concordant with the presence of the mitochondria, outside of loss in some lineages such as apicomplexa. Remarkably, human Ptrh1 was recently described as a release factor in a variant RQC pathway where the nascent peptide chains had not yet been ubiquitinated [61]. The authors report that the tRNAs attached to these chains are not “fixed” to the P site, consistent with the bacterial Pth enzymes acting on translation intermediates that dissociate from the ribosomal active sites. The Pth superfamily belongs to a vast monophyletic assemblage of enzymes, and additionally includes several families with peptidase, phosphorylase, and dioxygenase activities. Another member of this assemblage, which is highly conserved in archaea, catalyzes a similar reaction, i.e., hydrolysis of the amino acyl bond of tRNAs mischarged with D-amino acids. This observation together with the fact that Pth appears to be one of the earliest branching members of this assemblage suggests that the hydrolysis of aminoacyl/peptidyl tRNA in contexts other than translation termination might have been an ancient function of this assemblage of enzymes. 

## 4. Conclusions

Termination is a universal step in the translation process. However, paradoxically, sequence and structure analysis indicate that unlike the factors mediating the key steps of amino acylation, peptide-bond formation and elongation, the peptidyl hydrolases and GTPases catalyzing translation termination are evolutionary distinct or unrelated in the two great divisions of life. This leads to the idea that the LUCA probably did not possess a dedicated protein-based apparatus for release of the peptidyl-tRNA during transcription termination, and that this function appears to have convergently evolved only after the bacterial and archaeo-eukaryotic lineages separated. Further, recent analyses of the RF-PH domains have identified several novel versions. The genomic contexts and experimental evidence hint at a diversity of contexts in which such release factors might be deployed beyond conventional translation termination. These appear to be key players in processes like ribosomal quality control and repair of ribosomes “jammed” by ribosome-targeting effectors deployed by biological conflict systems. These investigations also open the door for understanding novel ribosome rescue pathways, including in the archaea, where little is known about specialized translation termination processes. Further, the RNA repair-coupled translation termination systems (e.g., prfH/RFH system) offer opportunities for developing novel biotechnological applications.

## Figures and Tables

**Figure 1 ijms-20-01981-f001:**
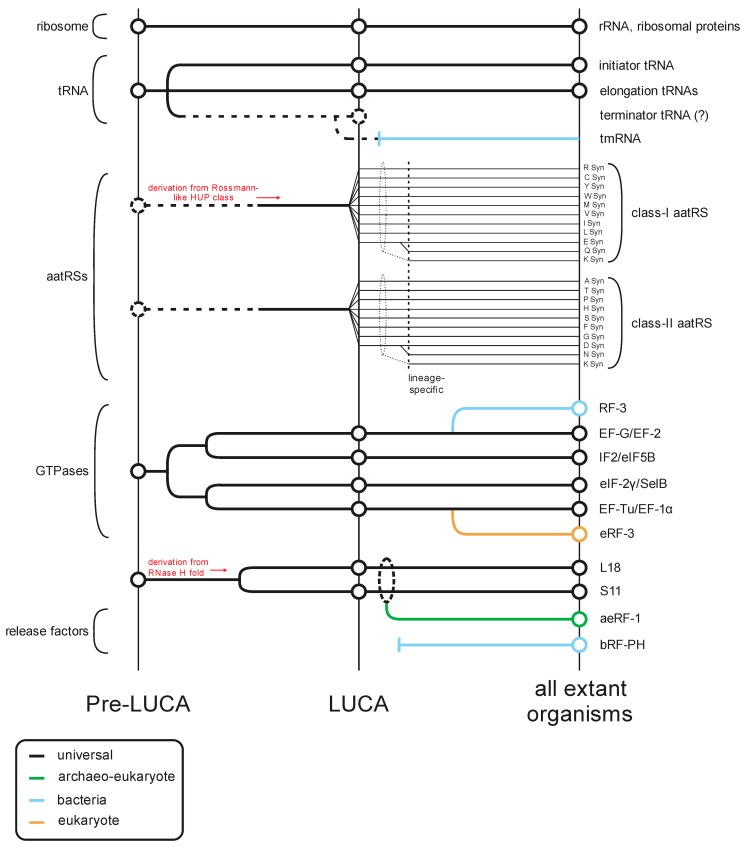
Overview of the evolutionary history of the translation apparatus. Key classes of protein and RNA contributors (general protein/RNA class labeled to the **left**, specific component names provided to the **right**) in translation are traced across several evolutionary epochs (labeled at the **bottom**) in this temporal diagram. Lines track evolutionary depth for components and are color-coded by phyletic distribution. Key derivation events are noted in red. Events where evolutionary depth are unclear are denoted with broken lines.

**Figure 2 ijms-20-01981-f002:**
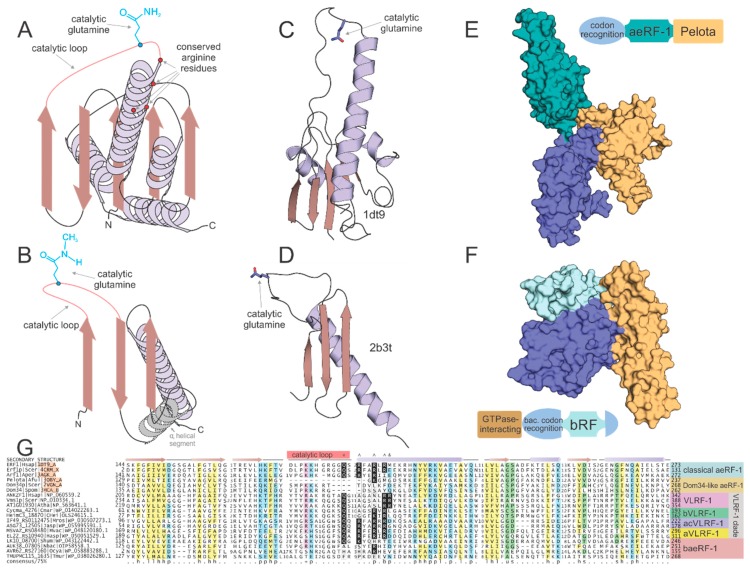
Structural and sequence overview of the catalytic domains of the two types of RF-PHs. (**A**,**B**) Topology diagrams of (**A**) aeRF-1 superfamily and (**B**) bRF-PH domain. (**C**,**D**) Cartoon rendering of solved crystal structures of known examples of (**C**) aeRF-1 superfamily and (**C**) bRF-PH domain. Protein DataBank id (PDBID) provided to right of structure. Coloring as in (**A**,**B**). Catalytic glutamine residue rendered as ball-and-stick. (**E**,**F**) Space-filled rendering of (**C**,**D**), while including full complement of domains found in the RF-PHs; positioning altered from (**C**,**D**) to emphasize proposed tRNA mimicry of the tri-domain release factors. Domain architectures provided adjacent to space-filled structural renderings, color-coded with structure to demarcate domains. (**G**) Multiple sequence alignment of aeRF-1 superfamily. Sequences are labeled to the left of the alignment and secondary structure is provided above the alignment. The catalytic loop region is marked with a pink box along secondary structure line and the catalytic glutamine residue position is marked with an asterisk above. Conserved positively-charged positions are marked with ‘^’. Conserved arginine position present only in eVLRF-1 family is marked with ampersand. Consensus line provided below alignment, coloring and abbreviations as follows: p, polar residue shaded in blue; ‘+’, positively-charged residue shaded in purple; h, hydrophobic residue shaded in yellow; l, aliphatic residue shaded in yellow; u, tiny residue shaded in green; s, small residue shaded in green; b, big residue.

**Figure 3 ijms-20-01981-f003:**
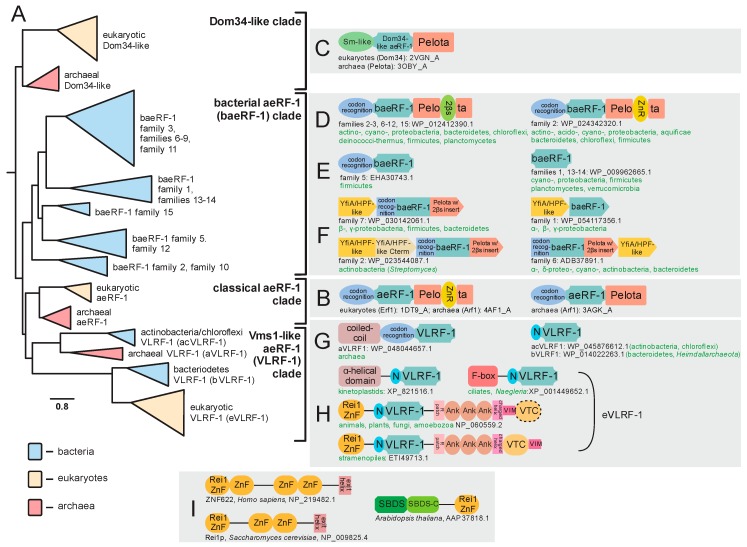
Evolution of aeRF-1 superfamily with known domain architectures and conserved gene neighborhoods. (**A**) Collapsed phylogenetic tree depicting relationships within the aeRF-1. (**B–I**) Notable domain architectures and conserved gene neighborhoods grouped according to clades. Individual domains of any given single protein architecture are depicted as discrete, colored shapes. In (**F**), individual genes in conserved gene neighborhoods are depicted as boxed arrows with the arrowhead pointing in the 3′ direction. Genes with multiple domains have their boxes split into distinct colors, one color for each discrete domain. Domain abbreviations: VTC, Vms1 Treble Clef; ZnR, zinc ribbon; 2βs, 2-β-strand insert domain; SBDS, Shwachman-Bodian-Diamond syndrome; N, distinctive N-terminal extension shared between certain families of the VLRF-1 clade.

**Figure 4 ijms-20-01981-f004:**
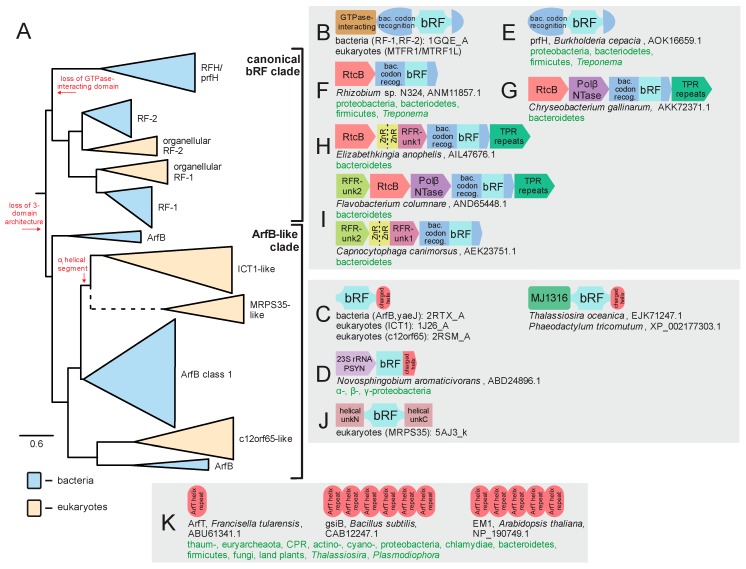
Evolution of bRF-PH domain with known domain architectures and conserved gene neighborhoods. (**A**) Collapsed phylogenetic tree depicting the radiation of the bRF-PH domains. Key transitions are denoted in red lettering along the tree. (**B–K**) Notable domain architectures and conserved gene neighborhoods. See Figure 3 legend for further descriptions and details. Abbreviations: PSYN, pseudouridine synthase; RFR, RFH and RtcB-associating unknown domains; bac. codon recog., bacterial codon recognition domain.

**Figure 5 ijms-20-01981-f005:**
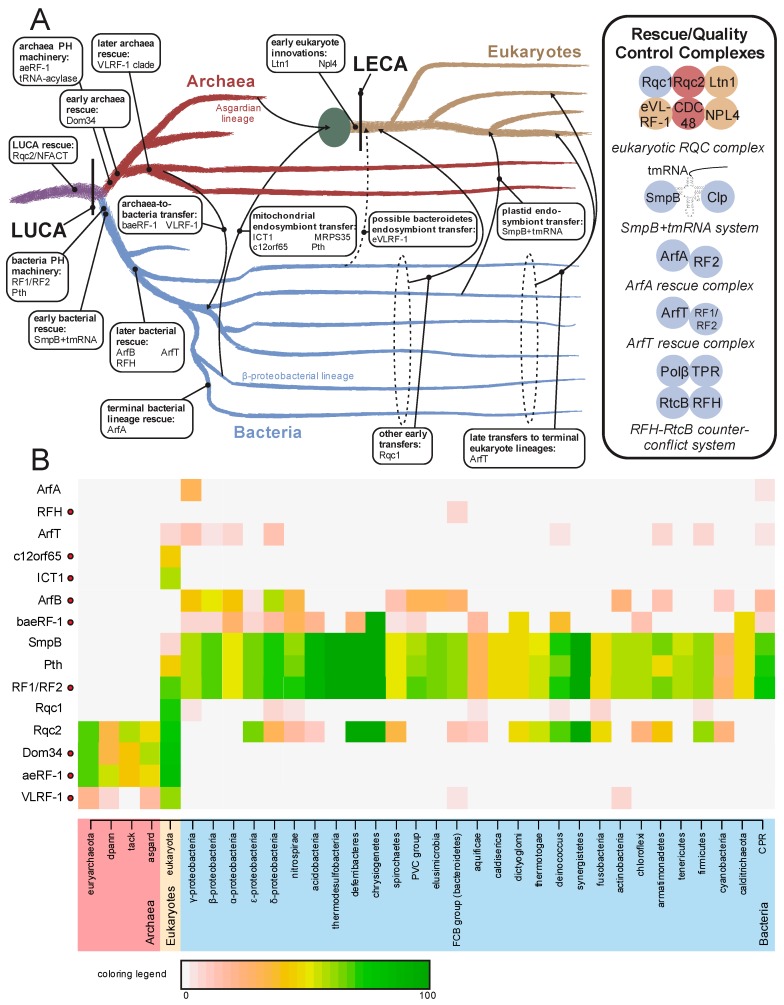
(**A**) Evolutionary origins and trajectories of release factors and ribosome rescue pathways, with factors listed at points of emergence/transfer. Transfer events are depicted with arrows and labeled; when timing is unclear the lines are broken. The mitochondrial endosymbiotic event is represented by a large green circle. Complexes formed by release/rescue factors are depicted to the right. Circles are color-coordinated with tree to represent the superkingdom of origin of a component. (**B**) Relative presence/absence for individual components of the ribosome quality control and rescue pathways. Presence/absence is colored by percentage of genomes within taxa which contain a given component. Names of taxa are colored by superkingdom. Components containing a RF-PH domain are marked to the right of their name by a red circle. Abbreviations: tack, Thaumarchaeota, Crenarchaeota, and Korarchaeota superphylum; dpann, Diapherotrites, Parvarchaeota, Aenigmarchaeota, Nanoarchaeota, and Nanohaloarchaea superphylum; PVC group, Chlamydiae, Lentisphaerae, Planctomycetes, and Verrucomicrobia bacterial superphylum; FCB, Fibrobacteres, Bacteroidetes, and Chlorobi superphylum; CPR, Candidate Phyla Radiation.

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
