# Peer review of "The Origin and Evolution of Release Factors: Implications for Translation Termination, Ribosome Rescue, and Quality Control Pathways"

_ijms, 2019, doi:10.3390/ijms20081981_

Reviewer 1 Report

In this extensive review, the authors summarize and aggregate current knowledge about ribosomes release factors, in particular their evolutionary roots and how this relates to today’s roles of the different proteins in the different kingdoms of life.

The authors do a very decent job structuring their findings, and likewise nicely illustrate them. They go into great details for each protein family and point out functional differences between prokaryotes and eukaryotes, and how this is reflected in the proteins’ structure.

1. I overall find this an elegant, comprehensive review, that more than merits publications.
2. One suggestion for improvement would be to provide a little more detail on ribosome recycling, which the authors only slightly touch upon.

3. Besides a few minor grammatical spelling issues, my main criticism at this point is the illustration. Most of the text elements within the figures are not readable, and despite this being in part due to the conversion into PDF format, the text will be too small for the reader. This in particular holds true for Figure 2G, 3A, 4A, 5A. I would strongly urge the authors to rearrange the figures to increase readability.
4. In addition, the lettering of Figure 3 seems odd.

Due to the fact that the illustrations prevent an in-depth review of their content, I am recommending to insist on a revised version of this manuscript.

Author Response

1. We appreciate the reviewer’s concise summary and kind words.

2. We have added a description of the evolutionary parallels of the recycling factors to the release factors in the introduction. This should provide a stronger background for the points in the text where ribosome recycling intersects with rescue and other pathways, as mentioned by the reviewer.
3. We have increased the font sizes in the sub-figures flagged by the reviewer to the maximal extend possible. We also agree with the reviewer that the supplied figures were unacceptably low in terms of their resolution. We hope the journal allows higher-resolution figures to be uploaded and incorporated in the manuscript.
4. We have attempted to keep the lettering in all figures as easy to follow as possible. Unfortunately, due to the tree topology and the order in which the different aeRF-1 release factors must be introduced in the text (i.e. describing well-characterized versions first), the lettering cannot proceed in a strict order which would follow typical top-to-bottom or right-to-left conventions.

Reviewer 2 Report

This manuscript by Burroughs and Arvin reviews the origin and evolution of release factors catalysing the release of the polypeptide from the ribosome. The manuscript describes the evolutionary driver of innovation in rescue pathways in respons to biological conflicts that target the ribosome. The manuscript is well written.

I only have some minor comments:

1. In my opinion the introduction would benefit from a short general section about translation, possibly with a simple figure.

2. Bad quality of the figures. The figures are well designed, but maybe as a consequence of compression to generate the pdf, the text is (almost) not readable.

3. The authors could expand a bit on the rescue of stalled ribosomes section

Author Response

We appreciate the reviewer’s assessment and kind comments.

1. We have altered the beginning of the introduction to more clearly overview the structure and mechanism of translation. We keep the focus of the introduction on the evolutionary conservation of the conserved nucleic acid and protein components of translation, consistent with the intended thrust of this review. The coupling of this revised introduction with the current Figure 1 should greatly clarify how termination fits into the greater scheme of translation, without necessitating a new figure of subfigure.

2. As described in the response to reviewer #1, we were unaware of how poorly-rendered the figures were in the version extended to reviewers. The figure versions we were made privy to during upload appeared adequate for review; however, we absolutely agree that the current state is unacceptable. We have increased the font sizes to the extent possible across all figures, but the key to addressing this issue is to work with the journal during upload to ensure that higher-resolution figures are uploaded and ultimately incorporated in the final manuscript.

3. We appreciate the reviewer raising this issue. We have introduced more thoroughly the kinds of conditions leading to ribosome stalling earlier in the paper at the introduction of the Dom34 release factor homolog. We have discussed this more extensively at the beginning of the section in question. We believe this better frames the conditions resulting in ribosome stalling and also the evolutionary forces driving the recruitment of RF-PH paralogs to the pathways combatting ribosome stalling.

Round  2

Reviewer 1 Report

The authors di a nice job addressing the suggestions in the previous review.

I believe the paper is suitable for publication, with the small exception that there are a few formatting (capitalization) issues in the references.

Author Response

We thank the reviewer for his comments. We have now modified to manuscript to correct the capitalization issues mentioned by the reviewer

Reviewer 2 Report

This review, entitled "The origin and evolution of release factors: implications for translation termination, ribosome rescue and quality control pathways" describes the diversification of release factor peptide hydrolases. 

The authors have addressed my comments in this revised version of the review.

Author Response

We thank the reviewer for the comments and have checked the manuscripts for spellings